# Investigation of Adhesive’s Material in Hermetic MEMS Package for Interfacial Crack between the Silver Epoxy and the Metal Lid during the Precondition Test

**DOI:** 10.3390/ma14195626

**Published:** 2021-09-27

**Authors:** Mei-Ling Wu, Jia-Shen Lan

**Affiliations:** Department of Mechanical and Electro-Mechanical Engineering, National Sun Yat-sen University, 70 Lien-Hai Rd., Kaohsiung 804201, Taiwan; d023020004@student.nsysu.edu.tw

**Keywords:** Micro-Electro-Mechanical Systems, moisture sensitivity level test, reflow process, finite element method, cohesive zone model, bonding strength, precondition test

## Abstract

A hermetic Micro-Electro-Mechanical Systems (MEMS) package with a metal lid is investigated to prevent lid-off failure and improve its reliability during the precondition test. While the MEMS package benefits from miniaturization and low cost, a hermetic version is highly sensitive to internal pressure caused by moisture penetration and the reflow process, thus affecting its reliability. In this research, the finite element method is applied to analyze the contact stress between the metal lid and the silver epoxy by applying the cohesive zone model (CZM). Moreover, the red dye penetration test is applied, revealing a microcrack at the metal lid/silver epoxy interface. Further analyses indicate that the crack is caused by internal pressure. According to the experimental testing and simulation results, the silver epoxy material, the curing process, the metal lid geometry, and the bonding layer contact area can enhance the bonding strength between the metal lid and the substrate.

## 1. Introduction

The MEMS package is widely used in automotive, medical, and consumer electronic devices for measuring the mechanical, thermal, optical, and magnetic phenomena. Therefore, it needs to be highly reliable to maintain its functionality and safety for several years. A metal lid is included in the MEMS package to provide protection from the external environment, while also reducing the cost, weight, and size, thus improving production efficiency. While the hermetic MEMS package offers considerable benefits, it also causes reliability issues for the lid bonding technology and hermeticity. Hsu et al. [1] investigated the characteristics of polymeric materials in the CMOS image sensor (CIS). They found that the hygroscopic swelling of polymer material is induced by absorbing the moisture in humid environments, which weakens the interfacial strength and causes delamination failure. To improve performance and reliability in RF-MEMS applications, Jeong et al. [2] developed novel wafer-level hermetic package technology. As a part of their study, low-temperature bonding technology was applied through gold/tin eutectic solder at the peripheral edge. The results revealed that thermal cycling, high-temperature storage, high-humidity storage, and a pressure cooker test failed to induce failure. Zhang et al. [3] investigated the bonding strength of a nanosilver sintered hermetic cavity with copper and silicon lids. Their results showed that the copper lid suffered delamination in the bonding layer, whereas the silicon lid exhibited great bonding quality. Farisi et al. [4] developed a low-temperature wafer-level hermetic packaging technology based on thermal compression bonding. Their analyses revealed that the bonding shear strength of the newly proposed technology exceeded 100 MPa and its leak rate was below 1.67 × 10^−15^ Pa·m^3^·s^−1^. Huang et al. [5] also developed low-cost and low-temperature hermetic technology based on a eutectic PbSn solder and Cr/Ni/Cu bonding pad. The bonding strengths of glass–glass, silicon–glass, and silicon–silicon pairs were measured at 4.5, 7, and 5.3 MPa, respectively. Jang et al. [6] proposed a diffusion-based governing equation to investigate the effects of polymer seal diffusion properties and geometries on the MEMS package performance. The numerical results revealed that both factors affected the lag time. Premachandran et al. [7] developed a wafer-level vacuum package with a wafer cap under the vacuum (1 mTorr). The package performance measured up to standard, evaluated via shear test and reliability tests. Jiang et al. [8] used a laser-assisted bonding method for a cavity-based package with a liquid crystal polymer (LCP). Their results showed that both silicon and glass substrates had high bonding quality. They also measured shear strength in the 20.8–26.1 MPa range, depending on the bonding assembly (glass–glass, silicon–glass, silicon–silicon, and silicon–package). Sandvand et al. [9] analyzed the bonding material stress in the MEMS pressure sensor for the glass-frit bonding process by conducting a finite element analysis. The authors observed microcracks at the outer perimeter of the glass-frit material due to the high stress levels induced by the thermal cycling test. As can be seen from the above, the bonding strength and the hermeticity of the MEMS package with the vacuum cavity have been thoroughly investigated. Nonetheless, the MEMS package reliability needs to be improved further for its greater use in automotive, medical, and consumer applications.

In the present research, the reliability of a hermetic MEMS package with a metal lid is evaluated through the precondition test. The hermetic MEMS package adopted for this purpose comprises of a ceramic substrate, two dies, and a metal lid. The metal lid and the ceramic substrate are bonded together with silver epoxy under atmospheric pressure, as the aim is to reduce cost and improve the fabrication process efficiency. However, as moisture inside the MEMS package cavity is a potential risk, during the reliability test, the aim is to prevent the lid-off and improve the bonding strength between the metal lid and the ceramic. Thus, in the analyses, focus is given to the curing process, the silver epoxy material, the metal lid geometry, and the bonding layer contact area.

## 2. Fabrication Process

The hermetic MEMS package with the metal lid used in this study was fabricated as shown in Figure 1. As can be seen from the diagram, once the stacked dies were bonded to the ceramic substrate with silver epoxy, the lid attachment and the precondition test were conducted.

(a)Lid attachment: The metal lid is attached to the ceramic substrate with silver epoxy, which is applied between the metal lid and the ceramic substrate of the peripheral MEMS package. To fully cure silver epoxy, it is exposed to the 175 °C temperature for 2 h. Subsequent evaluations confirm that silver epoxy fully adheres with the metal lid and the ceramic substrate.(b)Moisture sensitivity level (MSL) 1 test: The MSL 1 test is carried out to determine the sensitivity level of the hermetic MEMS package under humid conditions. For this purpose, the hermetic MEMS package is exposed to high humidity and high temperature (85 °C/85% RH). When the moisture penetrates into the MEMS package cavity via silver epoxy, it weakens the metal lid/silver epoxy and the ceramic substrate/silver epoxy bonding strength.(c)Reflow process: During the reflow process, the moisture concentration inside the MEMS package causes damage to the metal lid/silver epoxy interface. When the MEMS package is exposed to the maximum temperature of 265 °C for three cycles, vapor pressure and thermal pressure are induced by the residual moisture in the hermetic cavity. Furthermore, thermal stress is generated at the metal lid/silver epoxy and the ceramic substrate/silver epoxy interface due to the coefficient of thermal expansion (CTE) mismatch.

## 3. Root Cause

As shown in Figure 2, the precondition test results in an interfacial crack on the exterior of the hermetic MEMS package. The crack extends from the exterior along the metal lid/silver epoxy interface. This causes a phenomenon known as “lid-off” indicating that the metal lid is separated from the ceramic substrate. In the MEMS package, lid-off failure occurs because upward force is applied on the metal lid. To determine its root cause in the precondition test, the experimental design shown in Figure 3 was adopted in this study. The shear test and the red ink penetration test were performed to record the results, which are denoted as Result A (only reflow), Result B (only MSL-1), and Result C (reflow and MSL-1).

The shear test and the red dye penetration test results are presented in Figure 4. As can be seen from Result A (only reflow), the red ink is located at the outside of the MEMS package, indicating that no cracks have occurred at the interface or in the silver epoxy under the reflow process. When the MEMS package is exposed to high humidity, the moisture penetrates into the silver epoxy. As the residual moisture weakens the metal lid/silver epoxy bonding strength, the red ink penetrates inside the silver epoxy and the package, as indicated by both Result B (only MSL-1) and Result C (MSL-1 and reflow). The maximum shear force also decreases as a result of moisture penetration. Result C further reveals that the metal lid has separated from the silver epoxy, as the moisture inside the MEMS package cavity vaporizes and generates vapor pressure during the reflow process. Hence, the vapor pressure and the thermal pressure have a potential to cause the lid-off.

## 4. The Shear Test in the Different Manufacture Condition

In the shear test, the thrust force is applied on the bottom side of the metal lid to remove it from the MEMS package. During this process, the maximum shear force is measured to determine the shear strength of both the metal lid and the silver epoxy. To investigate the influence of the internal pressure on the likelihood of lid-off failure, a hermetic MEMS package with vent hole was designed, as shown in Figure 5. The vent hole was drilled at the corner and the top of the metal lid, allowing the internal pressure to be released during the reflow process. To analyze the maximum shear force under different manufacturing conditions, the hermetic MEMS package with a vent hole was compared to that without a vent hole, as shown in Figure 6 and Figure 7. As can be seen from Figure 6, the maximum shear force of the hermetic MEMS package without a vent hole after the precondition test (1.39 kgf) is lower than that measured for the hermetic MEMS package without a vent hole before the precondition test (3.18 kgf). However, the maximum shear force of hermetic MEMS package with a vent hole measured before the precondition test is similar to that obtained after the test. These results indicate that the internal pressure is a critical factor for lid-off failure under the reflow process.

To improve the maximum shear force of the hermetic MEMS package, additional tests were performed while controlling for the degree of curing and the pre-heat conditions, as these factors affect the material characteristics and the interfacial contact strength of the silver epoxy. To obtain fully cured and incompletely cured epoxy, the following conditions were respectively applied: 175 °C/1 h + 190 °C/1 h and 175 °C/1 h. As fully cured silver epoxy is harder and has a higher Young’s modulus, its maximum shear force is higher than that of the partially cured epoxy. Our analyses further indicate that when the fully cured epoxy is used in the hermetic MEMS package with a vent hole and the pre-heat (110 °C/0.5 h) step is performed, the maximum shear force increases by about 60% relative to the partially cured epoxy. In addition, when the fully cured epoxy is used in the hermetic MEMS package without a vent hole, the maximum shear force increases by about 48.5% after pre-heating. By observing the experimental testing results, the pre-heat does not have an effect on the hermetic MEMS package with a vent hole. The pre-heat condition can relieve the internal pressure applied on the metal lid without a vent hole in the curing process.

## 5. Finite Element Method

The metal lid detaches from the ceramic substrate because of internal pressure during the reflow process. To analyze the stress and the deformation of the hermetic MEMS package with a metal lid under the reflow process, a finite element model was adopted by using ANSYS APDL. Specifically, the CZM method was used to calculate the contact stress at the metal lid/silver epoxy interface, which were denoted as contact and target elements. The MEMS package structure comprised of stacked dies, a die attach, a ceramic substrate, a metal ring, a silver epoxy, and lid metal, as shown in Figure 8. For modeling this structure, a two-dimensional finite element model with quadratic elements was established and was matched with scanning electron microscope (SEM) cross-section images. The material properties of the finite element model are presented in Table 1. During modeling, internal pressure was applied on the inside surface of the metal lid to simulate air pressure and vapor pressure in the cavity under the reflow process.

The internal pressure inside the cavity can be obtained by using the ideal gas equation, as the following equations:(1)PV=nRT
(2)$$P_{Internal}=P_{Air}+P_{Vapor}$$
(3)$$P_{Air\_265^{\circ }C}=P_{Air\_25^{\circ }C}\cdot \frac{T_{265^{\circ }C}}{T_{25^{\circ }C}}$$
(4)$$\begin{array}{cc} P_{Vapor\_265^{\circ }C} & =P_{85^{\circ }C/85\%\mathrm{RH}}\cdot \frac{T_{265^{\circ }C}}{T_{85^{\circ }C}}\\ & =(0.85\;\cdot \;P_{85^{\circ }C,SAT})\cdot \frac{T_{265^{\circ }C}}{T_{85^{\circ }C}} \end{array}$$
where *P* is the pressure, *V* is the volume, *n* is the number of moles of gas, *R* is the idea gas constant (8.317 $J\cdot mol^{-1}\cdot K^{-1}$), *T* is the absolute temperature, *P_Internal_* is the internal pressure, *P_Air_* is the air pressure, *P_Vapor_* is the vapor pressure, *P_Air_25_**_°C_* is the air pressure at 25 °C, *P_85_**_°C,SAT_* is the saturated vapor pressure at 85 °C, and *RH* is the relative humidity.

The findings pertaining to the hermetic MEMS package with and without a vent hole were once again contrasted to investigate the contact stress and SEM observations, as shown in Figure 9. For evaluating the crack location, contact stress was defined as normal interface stress. In the hermetic MEMS package without a vent hole, the highest contact stress was located at the bottom of the metal lid. The fracture occurred at the same location during experimental testing. In the hermetic MEMS package with a vent hole, the contact stress was negligible and no fracture could be observed on the SEM images.

## 6. Optimization

### 6.1. One Factor Design

To decrease the contact stress at the metal lid/silver epoxy interface, one factor design was performed, considering lid thickness, connecting angle, epoxy height, lid height, lid size, and substrate height as factors, as presented in Figure 10. These design factors were chosen to evaluate the contact area effect, the lid geometry effect, and the material property effect, as indicated in Table 2. The lid thickness, the connecting angle, and the epoxy height are considered to exhibit the contact area effect since these factors are related to the contact interface area. The lid geometry not only affects the lid size but also has an influence on the cavity volume. Therefore, a lid of greater size would have higher contact stress due to withstanding higher internal pressure.

The one factor designs for the contact area, the lid geometry, and the material property effects are shown in Figure 11, Figure 12 and Figure 13, respectively. According to the assessments related to the contact area effect, greater lid thickness, and epoxy height, and a lower connecting angle reduce the contact stress by increasing the contact interface area. The lowest contact stress (2.45 MPa) is obtained with the connecting angle of 50°. According to the lid geometry effect, the lid size is sensitive to contact stress because the force induced by internal pressure is based on the lid size. Specifically, the contact stress increases from 1.36 to 7.15 MPa when the lid size increases from 2.5 to 6.1 mm. Finally, the results related to the material property effect indicate that the Young’s modulus of the silver epoxy and the lid do not exert significant changes on contact stress. Thus, even though Young’s modulus of the silver epoxy is not the critical factor, the moisture absorption, shear strength, and the material curing characteristics are important for the contact stress.

### 6.2. Responsed Surface Method

The response surface method was also adopted to establish the relationship between the factors that are most influential on contact stress, as shown in Figure 14 and Figure 15. As lid thickness, lid size, and the connecting angle are the critical factors for contact stress, their values were considered when interpreting the response surface results. By observing the relationship between lid thickness and lid size, it is evident that the slope of lid size is linear and is greater than the lid thickness. While the impact of lid thickness on contact stress is low, the curve flattens with increasing lid thickness. These results indicate that lid size is more significant than lid thickness. In addition, according to the response surface results based on the relationship between the connecting angle and the lid size, both factors exhibit linear distribution. Thus, for improving the contact stress, lid thickness should be increased, while its size and the connecting angle should be reduced.

## 7. Conclusions

In the research reported here, a hermetic MEMS package with a metal lid was designed and its reliability was tested. During the precondition test, lid-off failure occurred because moisture weakens the interfacial bonding strength and increases the internal pressure during the reflow process. The finite element method, which was adopted to simulate the contact stress of the metal lid/silver epoxy interface and the deformation of the metal lid, revealed that lid thickness, lid size, and the connecting angle are the critical factors for the contact stress. Although the Young’s modulus of the silver epoxy is not the critical factor, the moisture absorption, shear strength, and material curing characteristics are important for the contact stress. In the experimental testing, the pre-heat step and fully curing the epoxy can enhance the maximum shear force by 60% and 48.5% under the precondition test. The findings further indicate that increasing the lid thickness, and decreasing the lid size and the connecting angle can decrease the contact stress, thus reducing the likelihood of lid-off failure under the precondition test.

## Figures and Tables

**Figure 1 materials-14-05626-f001:**
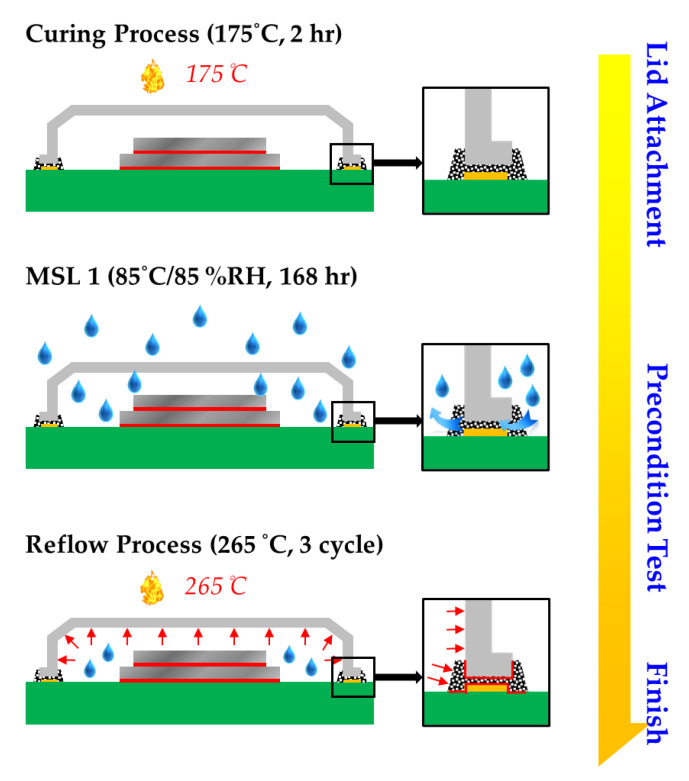
The fabrication process of MEMS package with metal lid.

**Figure 2 materials-14-05626-f002:**
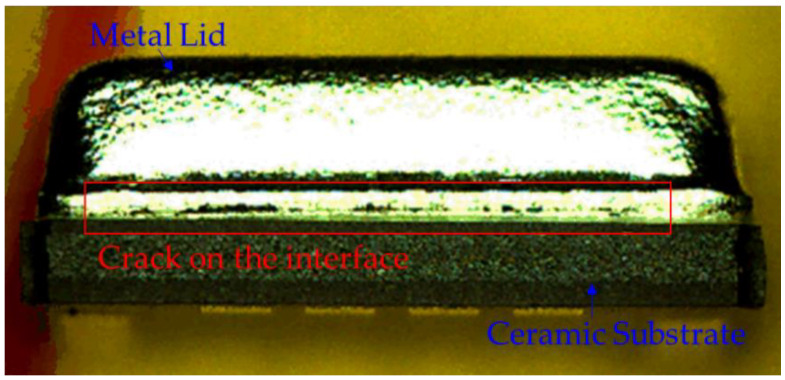
The crack occurred between the metal lid and the silver epoxy in the hermetic MEMS package.

**Figure 3 materials-14-05626-f003:**
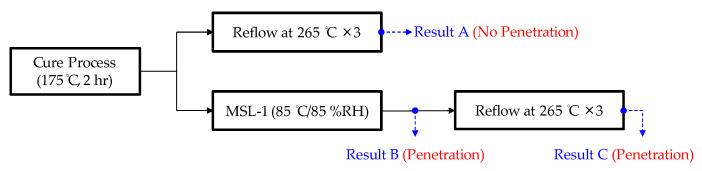
The process flow of investigating root cause.

**Figure 4 materials-14-05626-f004:**
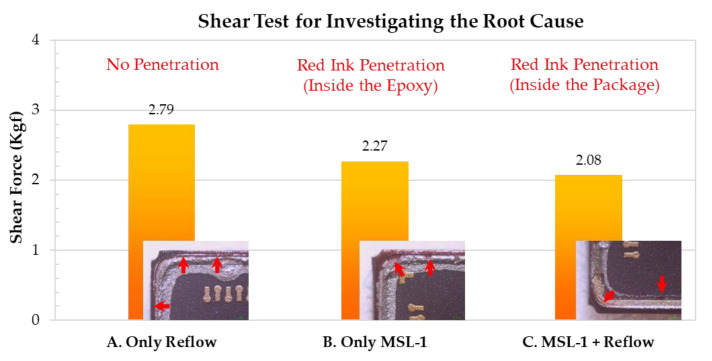
The shear test and red dye penetration test for investigating the root cause.

**Figure 5 materials-14-05626-f005:**
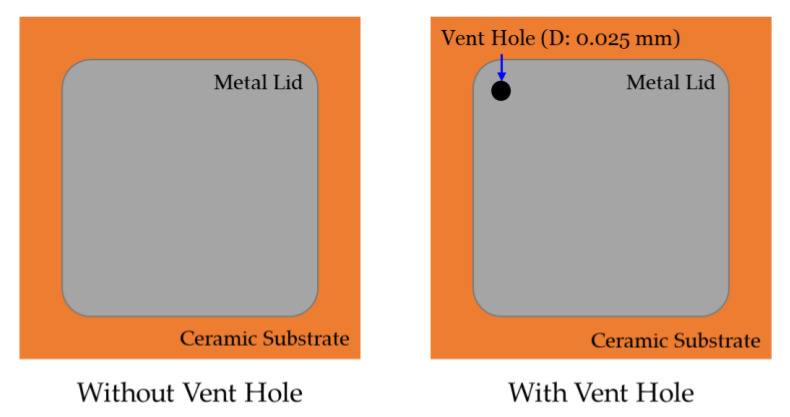
The schematic diagram of the hermetic MEMS package with a vent hole and without a vent hole.

**Figure 6 materials-14-05626-f006:**
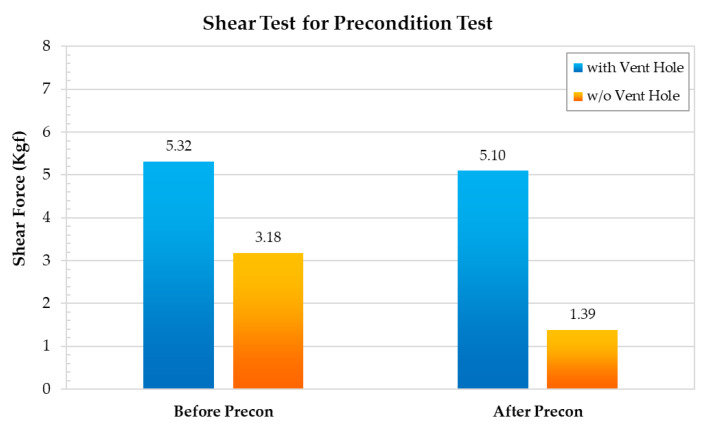
The shear test before and after the precondition test comparison.

**Figure 7 materials-14-05626-f007:**
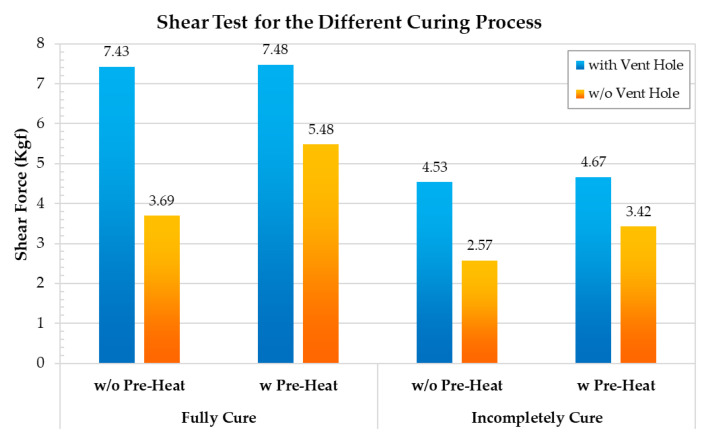
The shear test for discussing the degree of curing and the pre-heat condition.

**Figure 8 materials-14-05626-f008:**
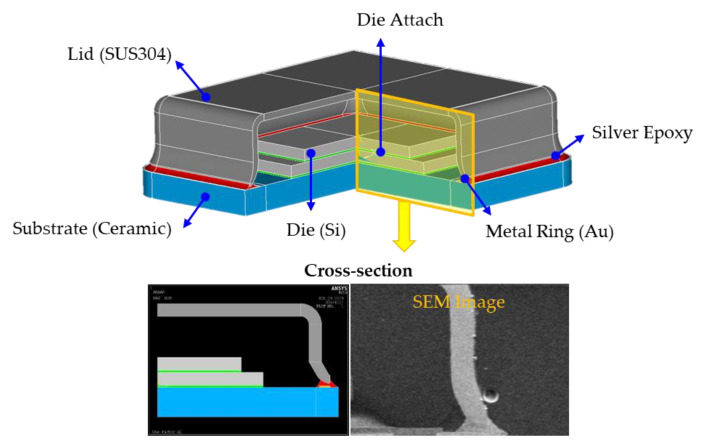
The finite element model of the hermetic MEMS package with metal lid.

**Figure 9 materials-14-05626-f009:**
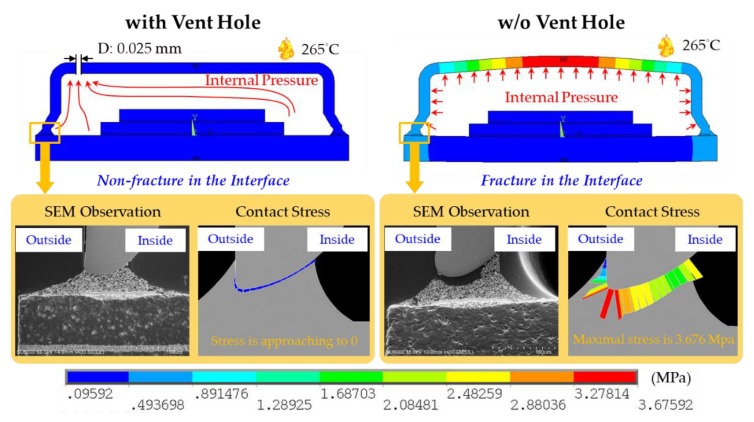
The contact stress (simulation result) and the SEM observation (experimental testing result) of the hermetic MEMS package with a vent hole and without a vent hole.

**Figure 10 materials-14-05626-f010:**
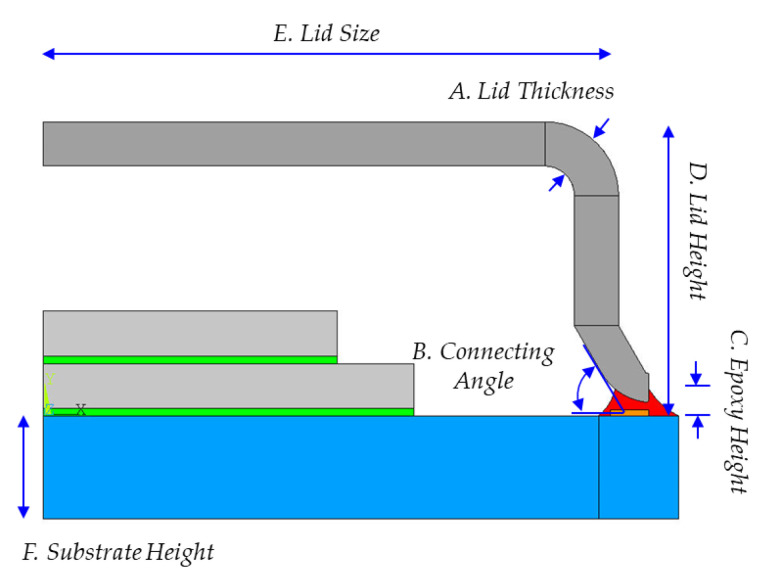
The schematic diagram of design factors.

**Figure 11 materials-14-05626-f011:**
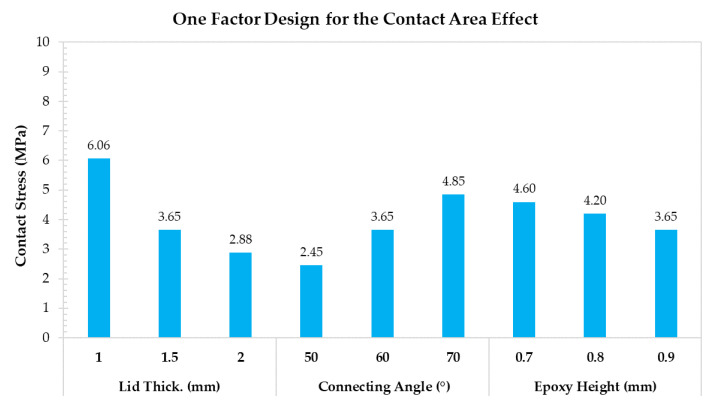
One factor design for discussing the contact area effect.

**Figure 12 materials-14-05626-f012:**
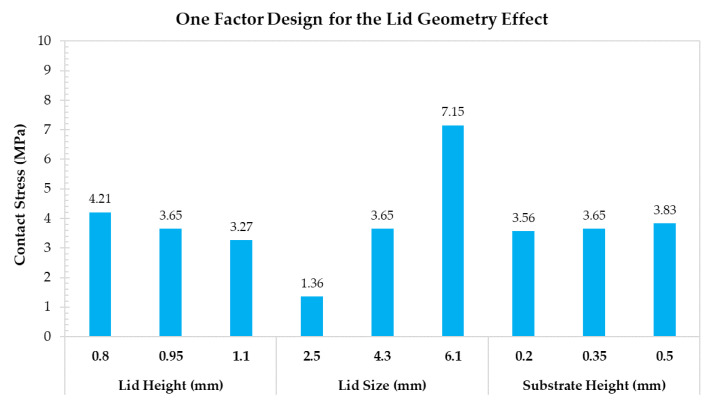
One factor design for discussing the lid geometry effect.

**Figure 13 materials-14-05626-f013:**
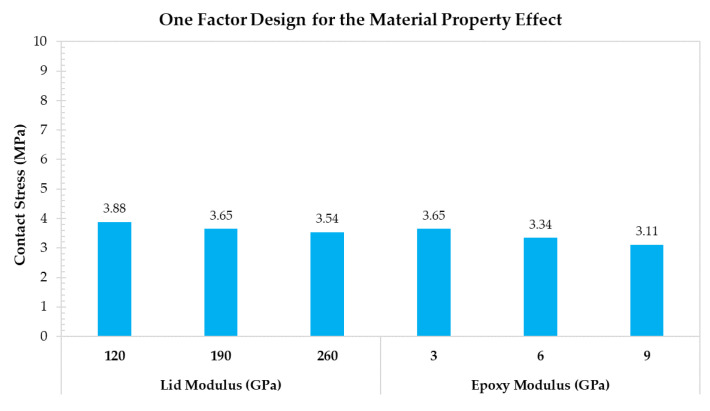
One factor design for discussing the material property effect.

**Figure 14 materials-14-05626-f014:**
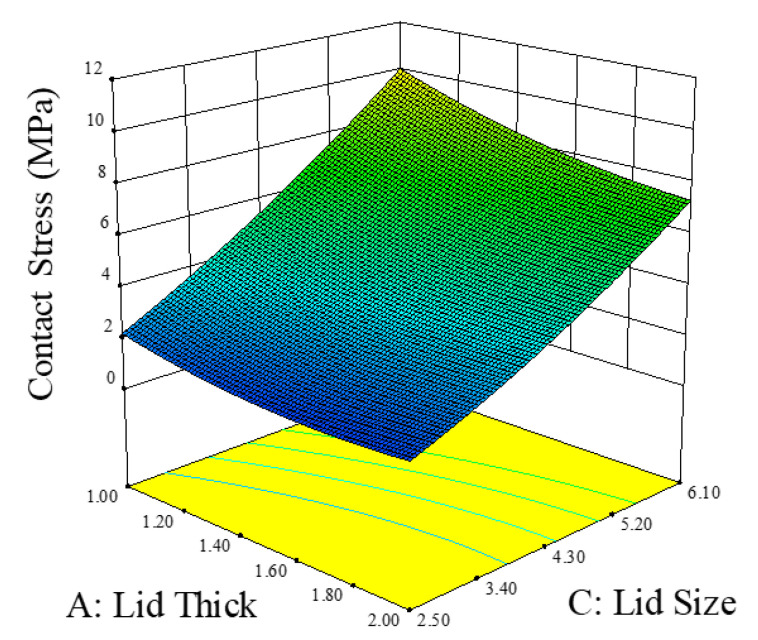
The response surface results for discussing the relationship between the lid thickness and the lid size.

**Figure 15 materials-14-05626-f015:**
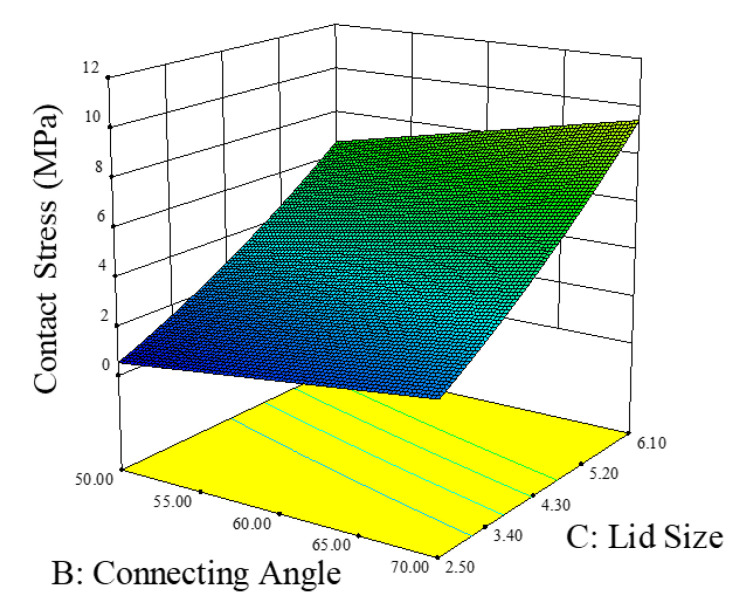
The response surface results for discussing the relationship between the connecting angle and the lid size.

**Table 1 materials-14-05626-t001:** The material properties of the finite element model.

	E (MPa)	ν	CTE (ppm/K)
**Die**	131 × 10^3^	0.27	2.8
**Lid**	190 × 10^3^	0.30	16.3
**Substrate**	310 × 10^3^	0.30	7.1
**Au**	77.2 × 10^3^	0.42	14.4
**Die Attach**	7.1 × 10^3^ @25 °C 0.6 × 10^3^ @260 °C	0.30	18 < 175 °C 35 > 175 °C
**Silver Epoxy**	3900@25 °C 2000@150 °C 300@250 °C	0.30	40 < 120 °C 150 > 120 °C

**Table 2 materials-14-05626-t002:** The design factors for analyzing the contact stress.

	Factors	Range
**The Contact Area Effect**	A. Lid thickness	0.1~0.2 (mm)
	B. Connecting angle	50~70 (°)
	C. Epoxy height	0.7~0.9 (mm)
**The Lid Geometry Effect**	D. Lid height	0.8~1.1 (mm)
	E. Lid size	2.5~6.1 (mm)
	F. Substrate height	0.2~0.5 (mm)
**The Material Property Effect**	G. Lid modulus	120~160 (GPa)
	H. Epoxy modulus	3~9 (GPa)

## Data Availability

Not applicable.

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
