# Peer review of "Investigation of Adhesive’s Material in Hermetic MEMS Package for Interfacial Crack between the Silver Epoxy and the Metal Lid during the Precondition Test"

_materials, 2021, doi:10.3390/ma14195626_

Round 1

Reviewer 1 Report

The authors present a paper titled “Investigation of adhesives material in hermetic MEMS package 2 for interfacial crack between the silver epoxy and the metal lid 3 during the precondition test”. There are a handful of specific technical concerns as below:

1) Did author verify the reliability of MEMS package with and without the vent hole?

2) Author should illustrate the name of the software for FEM analysis.

3) Did author characterize the MEMS package for the reliability to verify the model as shown in Fig. 10-13.   

Author Response

We thank the reviewers for their constructive criticism of and time spent analyzing this manuscript.  The responses and explanations related to their comments are listed below,thank you very much!

Reviewer 2 Report

Figure 10 is defining the «Lid height» as the distance from top substrate to top of lid. This do not agree with Table 2 where “Lid height” is in the range of 0.8 to 1.1 mm, and “Lid thickness” in Figure 12 is between 1.0 and 2.0 mm.

Line 222: “Therefore, a lid of greater size would be capable of withstanding higher internal pressure.” This statement must be wrong, as is also shown in Figure 12 (and also stated on line 280)

 Eq. (4) assumes that all the water present is in vapor form after the 85/85 MSL (precondition). The calculated pressure contribution from the water vapor do not allow for any absorbed water within the cavity volume (including silver epoxy) as a result of this humidity exposure.

Line 81: Full curing is given by 175 °C 81 temperature for 2 hours. In line 161 full curing is described as 175 °C/1 h + 190 °C/1 h. What is the correct?

Line 81:  Incomplete curing is described as 175 °C/1 h. Due to the asymptotic curing process, and the statement that 175 °C/2 h) is sufficient for full curing, we do not expect large difference in curing degree between these two processes. The large differences observed in Figure 7 therefore seems strange, and definitely deserve some explanation.

The effect of the TCE mismatch between the lid (stainless steel) and the substrate (ceramic) seems not to be considered in the FE simulations.

Line 134: “In the shear test, the pull force is applied on the bottom side of metal lid”. Most likely push.

Line 142: The expression “shear force” should probably be replaced by “maximum shear strength”. (This expression is repeatedly used in the document)

The effect of “Pre-Heat” is clearly shown for the test samples without vent hole (Figure 7). The authors should attempt to provide some explanation.

The geometry of the FE model is not sufficiently described. In particular, the interface region between lid, silver epoxy and ceramic seems to be very critical for the FEM results.

Author Response

(The authors gave the same response as above.)

Round 2

Reviewer 1 Report

Most of my comments from previous revision are answered and ready to publish.